# Bacteriostatic Poly Ethylene Glycol Plasma Coatings for Orthodontic Titanium Mini-Implants

**DOI:** 10.3390/ma15217487

**Published:** 2022-10-25

**Authors:** Juan Carlos Rodriguez-Fernandez, Francisco Pastor, Jose Maria Barrera Mora, Aritza Brizuela, Andreu Puigdollers, Eduardo Espinar, F. Javier Gil

**Affiliations:** 1Dept. Ortodoncia, Facultad de Odontología, Universidad de Sevilla, Avicena s/n, 41009 Sevilla, Spain; 2Facultad de Odontología, Universidad Europea Miguel de Cervantes, C/del Padre Julio Chevalier 2, 47012 Valladolid, Spain; 3Dept. Ortodoncia, Facultad de Odontología, Universidad Internacional de Catalunya, Josep Trueta s/n, Sant Cugat del Vallés, 08195 Barcelona, Spain; 4Bioengineering Institute of Technology, Facultad de Medicia y Ciencias de la Salud, Universidad Internacional de Catalunya, Josep Trueta s/n, Sant Cugat del Vallés, 08195 Barcelona, Spain

**Keywords:** bacteriostatic behavior, mini-implants, poly ethylene glycol, titanium, orthodontics

## Abstract

Titanium mini-implants are used as anchorage for orthodontic tooth movements. However, these implants present problems due to the infection of surrounding tissues. The aim of this work was to obtain a polyethylene glycol (PEG) layer by plasma in order to achieve a bacteriostatic surface. Titanium surfaces were activated by argon plasma and, after, by PEG plasma with different powers (100, 150 and 200 W) for 30 and 60 min. The roughness was determined by white light interferometer microscopy and the wettability was determined by the contact angle technique. Surface chemical compositions were characterized by X-ray photoelectron spectroscopy (XPS) and cytocompatibility and cell adhesion studies were performed with fibroblast (hFFs) and osteoblast (SAOS-2) cells. Bacterial cultures with *Spectrococcus Sanguinis* and *Lactobacillus Salivarius* were performed, and bacterial colonization was determined. The results showed that plasma treatments do not affect the roughness. Plasma makes the surfaces more hydrophilic by decreasing the contact angles from 64.2° for titanium to 5.2° for argon-activated titanium, with values ranging from 12° to 25° for the different PEG treatments. The plasma has two effects: the cleaning of the surface and the formation of the PEG layer. The biocompatibility results were, for all cases, higher than 80%. The polymerization treatment with PEG reduced the adhesion of hFFs from 7000 to 6000 and, for SAOS-2, from 14,000 to 6500, for pure titanium and those treated with PEG, respectively. Bacterial adhesion was also reduced from 600 to 300 CFU/mm^2^ for Spetrococcuns Sanguinis and from 10,000 to 900 CFU/mm^2^ for *Lactobacillus Salivarius*. The best bacteriostatic treatment corresponded to PEG at 100 W and 30 s. As a consequence, the PEG coating would significantly prevent the formation of bacterial biofilm on the surface of titanium mini-implants.

## 1. Introduction

Orthodontic treatment requires a balance in the orthodontic biomechanics. Anchorage control plays the main role of the orthodontic forces. This anchorage control is fundamental to successful orthodontic treatment. Several techniques that reinforce anchorage have been used in orthodontic therapies for multi-bracket or for aesthetic aligners. For both, additional anchorage supports are often needed to support the anchoring teeth, whereas intraoral aids to reinforce the anchorage are well-accepted due to the high loads, and extraoral systems that lack comfort, such as headgear, are often abandoned by patients. Titanium dental implants, because of their excellent capacity in osseointegration, provide an important rigid stability for bone–implant anchorage and serve as the best intraoral anchorage devices [1,2,3]. Since the last decade of the last century, unlike the osseointegrated dental implants, removable titanium mini-implants have been extensively used to provide excellent bone anchorage that resists high-orthodontic forces. These mini-implants are easy to insert and remove. Mini-implants, which were originally simply surgical miniscrews, have been developed and optimized to apply to many orthodontic strategies. Most importantly, given their small size, they can be placed in the alveolar bone of adjacent teeth without damaging roots [1,2,3,4]. The applications of mini-implants in orthodontics are highly versatile as can be seen in Figure 1.

However, the biggest problem encountered in the dental clinic is the bacterial colonization that can be created on the titanium surface. Biofilm causes gingival inflammation and bone loss and significantly reduces the attachment to the bone. In addition, inflammation of the gingival tissue around the head of the mini-implant is a risk factor for its stability [6,7,8]. In Figure 2, titanium mini-implants colonized by bacteria can be observed. Infections related to titanium mini-implants are difficult to treat since bacterial adhesion often leads to the formation of a biofilm, which is a multi-species community embedded in a polysaccharide extracellular matrix produced by the bacteria. The biofilm protects the bacteria community against the immune response and provides them with resistance to antibiotic treatments [7,8,9].

One of the best known antifouling polymers is poly (ethylene glycol) (PEG) [10,11,12]. PEG molecular chains are believed to resist protein adsorption by two mechanisms: steric repulsion due to the surface tension of the PEG when adsorbed by the titanium substrate and the barrier action created by the structured water associated with the PEG [13,14]. Many different approaches have been used to immobilize PEG on the biomaterials surface: self-assembly, physisorption, silanization, electropolymerization or plasma polymerization, among others [15,16,17,18,19,20,21]. However, though plasma polymerization has been extensively used on polymeric surfaces [21,22] to obtain PEG-like coatings, to the best of our knowledge, it has not been used on titanium surfaces.

In addition to PEG treatments, one of the most promising solutions used to induce the bactericidal character of titanium has been TiO_2_ nanotube formation treatments. Antibiotics and other drugs can be incorporated into these nanotubes to aid osseointegration and inhibit bacterial colonization [23,24]. In addition, in silico studies would be desirable to determine the influence of the mini-implant designs [25,26,27] and to determine the benefit of increasing the mechanical properties of the mini-implant with the Ti6Al4V alloy [28].

In this work, PEG coatings have been applied to titanium mini-implants for orthodontics. This contribution confirms the possibility of having bacteriostatic coatings on orthodontic mini implants. At present, these mini implants can suffer bacterial colonization, leading to orthodontic anchorage failure. Achieving a coating that inhibits the formation of biofilm is of great clinical interest. In this research, we intend to both study the possibility of obtaining a PEG coating and characterize its topographical and wettability properties, as well as the cellular and microbiological response, to determine if it can be a promising treatment for orthodontic mini-implants. 

## 2. Materials and Methods

Eighty cp-titanium mini-implants of grade 3 (HDC^®^ ·M, Mineapolis, MN, USA) that were 2 mm in diameter and 9 mm in length were used for surface characterization and biological and microbiological studies (Figure 3).

In Figure 4, we can see a scheme of the research carried out with the titanium mini-implants, the preparation of the samples coated with PEG plasma polymerization and the characterization carried out.

The methodology of plasma activation process for Ti surfaces was carried out as explained by Buxadera et al. [10]. These samples (n = 30) were treated by radio frequency low-pressure plasma apparatus (Plasma System Femto, Diener, Germany) using 13.52 Hz for 10 min treatment, with argon at a pressure of 0.40 mbar (Figure 5). This treatment was realized to activate the titanium surface and, for this reason, was carried out by argon nonpolymerizing gas. Plasma polymerization was performed right after plasma activation in the same reactor without breaking the vacuum. The polymerization precursor was tetra (ethylene glycol) dimethyl ether (tetraglyme, Sigma Aldrich, Sant Louis, MO, USA) introduced by bubbling argon in the reactor, and the parameters used were 100 W, 0.40 mbar and 1 h, according to a previous work [21]. The process was performed in pulsed mode with t_on_ = 20 µs and t_off_ = 20 ms. In Table 1, the different conditions of the plasma polymerization can be observed. Five samples were used for each treatment (power peak and time).

Roughness was determined by means of a white light interferometer microscopy (Wyko NT1100, Veeco, New York, NY, USA). Four samples for each treatment were analyzed and the measurements were realized in five surfaces to evaluate the Sa and Pc parameters. Data analysis was performed with Wyko Vision 232TM software (Veeco, New York, NY, USA). Based on previous tests, the following cut-off values were applied: λc = 0.8 mm and λc = 0.25 mm for control surfaces [29,30,31].

The values of the contact angle (SCA) (Contact Angle System OCA15plus, Dataphysics, Filderstadt, Germany) was carried out through the sessile drop method. The tests were realized at 25 °C in an environmental PMMA chamber that was saturated with the study liquid for three samples for each condition. The SCAs were determined with ultra-pure distilled water. At least three measurements were carried out with three different samples in each series. The contact angle measurements were performed with a contact angle video-based system and analyzed with SCA20 software [32,33].

X-ray photoelectron spectroscopy (XPS) was tested in ultra-high vacuum (5.0 × 10^−9^ mbar) with an XR50 Mg anode source operating at 150 W and a Phoibos 150 MCD-9 detector (D8 advance, SPECS Surface Nano Analysis GmbH, Berlin, Germany). C 1 s peak was used as a reference. As a reference used to compare the XPS results, the theoretical atomic composition of a PEG-amine of molecular weight 1500 g/mol was calculated by counting the number of atoms present in each polymeric chain [34,35]. These values were labeled as theoretical PEG. Three samples for each treatment were analyzed by XPS. 

Cytotoxic effects of control and PEG-coated surfaces were analyzed following ISO 10993-5 standard on human foreskin fibroblasts (hFFs, Merck Millipore Corporation, Bedford, MA, USA) as previously reported [16,21,36]. Ten samples for each treatment were studied. Extracts of the samples at concentrations of 1:1, 1:2, 1:10, 1:100 and 1:1000 were prepared by immersing the samples in Dulbecco’s modified Eagle medium (DMEM, Invitrogen, Carlsbad, CA, USA). A total of 5000 cells/well on a 96-well tissue culture polystyrene dish were in contact with the eluents for 24 h and then lysed with mammalian protein extraction reagent (mPER, Thermo Scientific, Waltham, MA, USA). Cell viability was measured by the activity of the enzyme lactate dehydrogenase (LDH) with a Cytotoxicity Detection Kit (Thermo Scientific, USA) as indicated by the supplier. 

Bacterial adhesion tests were analyzed with *Spectrococcus Sanguinis* CCUG 15915 (Culture Collection University of Göteborg (CCUG), Sweden) and *Lactobacillus Salivarius* CECT 101 (Colección Española de Cultivos Tipo, Valencia, Spain). Both cultures were incubated from three colonies overnight at 37 °C before the assays in Brain-Heart Infusion (BHI, Sharlab SL, Barcelona, Spain) using ten samples for each condition. Afterwards, bacteria suspensions were diluted to an absorbance of 0.20 ± 0.01 at 600 nm using a Laxco MicroSpek DSM-Cuvette Cell Density Meter (Cole Parmer, Vernon Hills, IL, USA), giving approximately 1 × 10^8^ colony forming units (CFUs)/mL. A total of 5 µL of the bacterial suspension was placed on top of the samples and left for 2 h at 37 °C. After, samples were cleaned twice with PBS. Adherent bacteria were detached by vortexing the disks for 5 min in 1 mL of PBS. Detached bacteria were then seeded using serial dilutions in BHI-agar plates [37,38]. The plates were then incubated overnight at 37 °C and the resulting CFUs were counted. Three samples for each condition were studied.

Cell-bacteria co-culture experiment was an adaptation from the study by Godoy-Gallardo et al. [39]. For cell adhesion studies, 2 × 10^4^ cells were seeded on each sample and left for 24 h at 37 °C. After 2 h at 37 °C, samples were washed three times in order to eliminate the non-attached bacteria, and hFFs in modified DMEM (DMEM with 2% BHI) at 2 × 10^4^ cells/sample were seeded and incubated for 24 h. 

The results were statistically studied by Student’s *t*-tests, one-way ANOVA tables and Turkey’s multiple comparison tests. Using this method can evaluate any statistically significant differences between the sample groups. The significance differences were when *p* < 0.05. The statistical study was realized by MinitabTM software (Minitab release 13.0, Minitab Inc., State College, PA, USA).

## 3. Results

Roughness studies were carried out on the mini-implants since, as is well known, roughness is a parameter that will affect bacterial colonization. The results of the original titanium (Ti), titanium activated by argon plasma (PA) and those treated with PEG polymerization plasma is shown in Table 2. From these results, no statistically significant differences can be observed in any case. The treatment does not affect the topography of the samples.

Wettability increased with the plasma activation, especially with argon (PA) treatments. This was realized by means of argon at a 100 W peak power from 64.2 ± 5.5° for the as-received titanium to 5.2 ± 1.2° treated with argon, showing an important super-hydrophilic character. The contact angles of the different treatments with PEG are shown in Table 3.

Statistically significant differences in the contact angle can be observed with the different peak power values applied, but no differences in treatment times are observed for each peak power applied.

The atomic concentration of the elements in the outer surface was recorded by XPS and is summarized in Table 4.

Cytocompatibility results demonstrated no decrease at any dilution when cultured with fibroblasts and osteoblasts (Figure 6 and Figure 7, respectively). All of the studied surfaces and the plasma polymerization conditions had cytocompatibility ratios over 70%. For both results, there are no significant statistical differences. The excellent biocompatibility of the treatments can be concluded.

Cell adhesion assays with hFFs (Figure 8) showed no difference between Ti, PA and PEG samples, whereas, for SAOS-2 (Figure 9), a slightly decrease was measured when the samples were polymerized.

Bacterial adhesion tests showed a decreased bacterial adhesion for all PEG samples, either for the *Spectrococcus sanguinis* and the *Lactobacillus salivarius* (Figure 10 and Figure 11, respectively). Ti samples and plasma-activated samples (PA) were used as controls. An increased bacterial adhesion was observed for the PA sample compared to Ti.

## 4. Discussion

In this study, we aimed to test if polyethylene glycol (PEG) coating would significantly prevent the formation of bacterial biofilm on the surface of titanium mini-implants. The roughness results show that the roughness is not affected by the plasma treatments. This fact is important since it is well known that roughness affects osseointegration levels, as well as bacterial proliferation and adhesion [40,41,42,43,44]. These results make it possible to optimize the treatments to obtain the optimum roughness, knowing that subsequent plasma treatments will not vary this important characteristic.

The wettability results indicate a very significant variation in the hydrophilic character when we activate the surface with argon. The variation in the contact angle from 64° to 5° makes the surface super-hydrophilic. Regarding the surface treatments with PEG, it can be observed in the results of Table 3 that, as we increase the peak power, the samples become less hydrophilic up to values of around 25° when applying 200 W. In addition, the PEG-treated samples reduce the contact angle by more than half with respect to the titanium of the mini-implant. This fact will be very important in order to avoid the bacterial adhesion. Furthermore, it can be observed that the application time has no significant influence on the wettability values [45,46].

The XPS results of Ti show oxygen on the surface, which accounts for the presence of titanium oxide and carbon, which may come from adsorbed contaminants on the surface. Regarding the XPS results of the PA samples, a decrease in the carbon amount on the titanium surface can be observed when comparing the untreated Ti to the plasma-activated one (Table 4) due to the cleaning effect of the treatment. When comparing the PEG samples with the PA ones, an increase in the intensity for the carbon peak can be observed, whereas the titanium peak decreases. The increase in the C/Ti ratio reflects the formation of a PEG-like coating on the surface. In addition to the surface cleaning effect of the plasma treatment, it also produces an activating effect on the surface to form bonds with polymers that can give functions, such as, in this case, producing the PEG anchor [21,47].

The PEG coatings have shown a decrease in the number of adhering cells, and especially in bacteria. A very significant decrease in bacterial adhesion can be clearly observed in the microbiological studies with the two bacterial strains, and therefore a significant bacteriostatic effect is shown. These results are in agreement with other works that show that PEG coatings produce a decrease in the adhesion of the BSA protein that favors bacterial colonization [10,21,48,49].

The results have shown a cytocompatibility higher than 80% in the fibroblastic and osteoblastic cells studied, and, regarding the differences in cell adhesion, although there is a slight decrease, there are no statistically significant differences. The results of the titanium treated with argon plasma show good biocompatibility and cell adhesion values, but do not offer good results in bacterial colonization. Of the plasma polymerization treatments, it can be seen that the peak power of 100 and 150 W and a time of 30 min offer the best results.

Marguier et al. [50] showed that the hydrophobic surface presents a clear decrease in bacterial adhesion and colonization, with areas completely free of bacteria regardless of the surrounding media, even after static long time culture or under hydrodynamic flow conditions. This fact may justify how superhydrophilic surfaces show poor bacterial behavior, as can be observed for argon-treated samples. These hydrophilic layers are able to adsorb proteins and bacteria on the surface. The lower the hydrophilicity, the lower the bacterial adhesion. In addition, changing the density and size of the surface nanostructure at the level of a 100 nm average height reduces bacterial colonization more than changing the wettability only in a non-clogging medium. As a consequence, PEG, due to its chemical configuration, causes a very important inhibition of bacterial affinity. The topography, wettability and heterogeneity of the coatings are specific to the manufacturing method, which, here, was through an atmospheric plasma deposition process. However, by considering the interplay between both topography, wettability and culture conditions, our statistical approach provides key insights for the design of low-wettable surfaces with bacterial anti-adhesive properties that might be obtained with other techniques.

## 5. Conclusions

Titanium surfaces were activated by argon plasma and then different PEG coatings were obtained using the same method. The results of the chemical composition of the titanium surface showed the cleaning effect of the plasma and the Ti and C ratio confirmed the appearance of the PEG layer. The plasma treatments do not affect the roughness, which is around 0.33 μm. It showed a higher hydrophilic capacity of plasma, ranging from 64.2° of as-received titanium to 5.2° when treated with Ar alone, and rose between 12 to 25° when treated with PEG. All PEG-coated samples showed a biocompatibility of more than 80% in fibroblast and osteoblastic cells. Polymerization treatment with PEG reduced the adhesion of hFFs from 7000 to 6000 and, for SAOS-2, from 14,000 to 6500 for pure titanium and those treated with PEG, respectively. Bacterial adhesion was also reduced from 600 to 300 CFU/mm^2^ for Spetrococcuns Sanguinis and from 10,000 to 900 CFU/mm^2^ for *Lactobacillus Salivarius*. The best bacteriostatic treatment corresponded to PEG at 100 w and 30 s. However, the titanium surface only activated by argon plasma caused an increase in bacterial colonization. Consequently, antifouling treatments using plasma-activated PEG coatings are a candidate for the introduction of orthodontic titanium micro-implants in order to prevent biofilm formation. This research should be extended with larger numbers of bacteria in order to see the bacteriostatic capacity that should be used to prevent biofilm formation. In vivo studies should be performed to determine adequate levels of osseointegration and the inhibition of bacterial colonization in induced infection.

## Figures and Tables

**Figure 1 materials-15-07487-f001:**
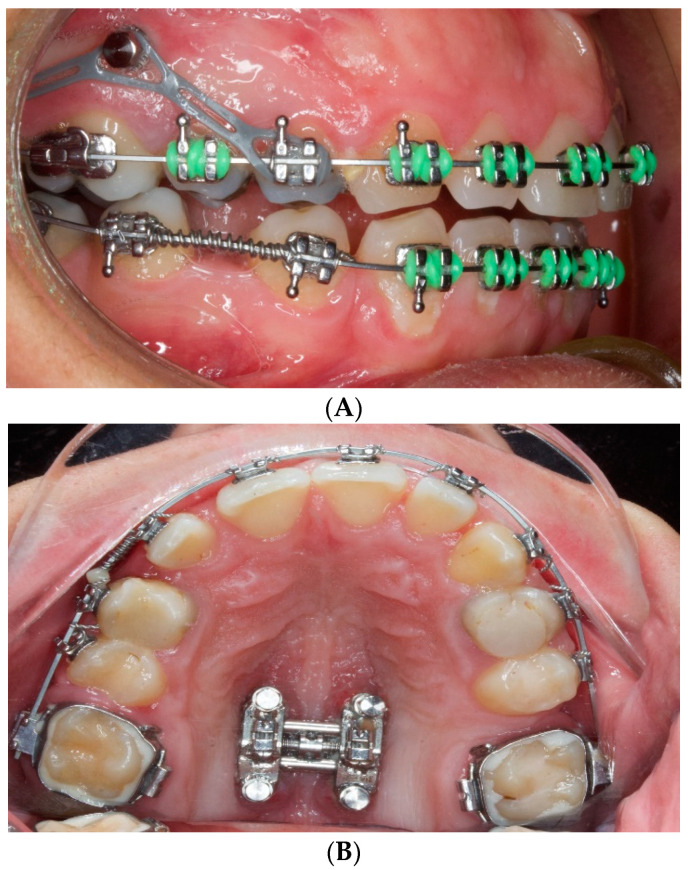
Mini-implants used in orthodontic therapies. (**A**) As anchorage for teeth movement. (**B**) To support a maxillary expander [5].

**Figure 2 materials-15-07487-f002:**
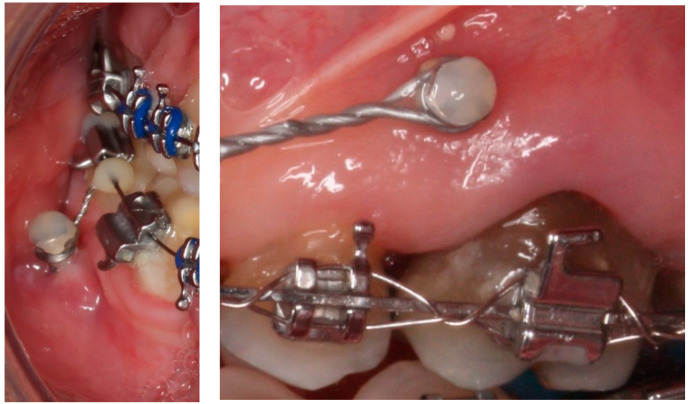
Gingival inflammation around the head of titanium orthodontic mini-implants [5].

**Figure 3 materials-15-07487-f003:**
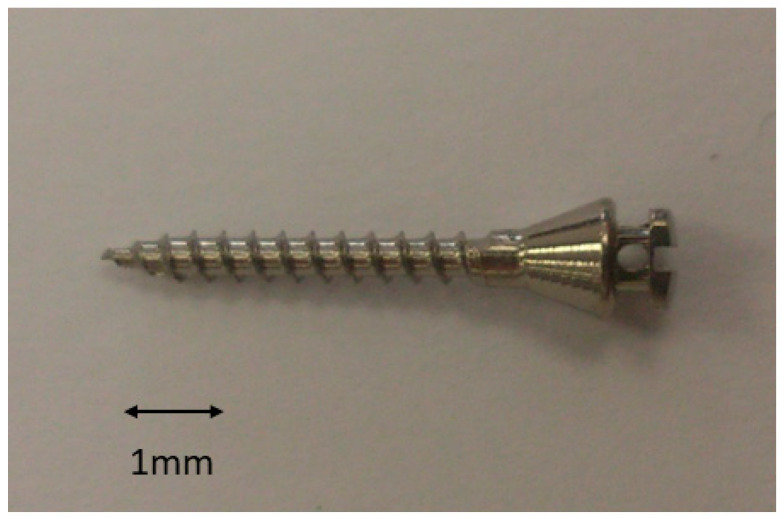
Orthodontic mini-implant used in this study.

**Figure 4 materials-15-07487-f004:**
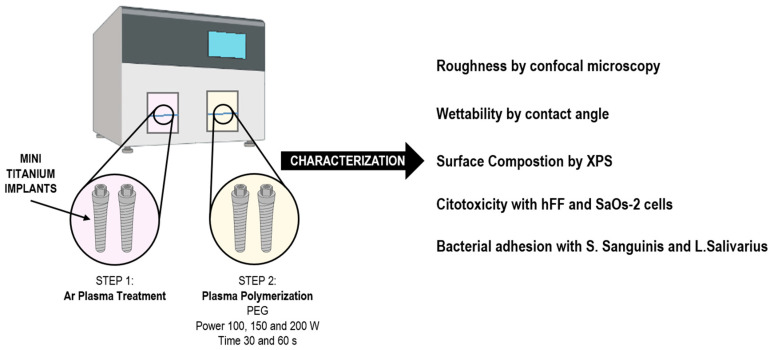
Schematic representation of the methodology used.

**Figure 5 materials-15-07487-f005:**
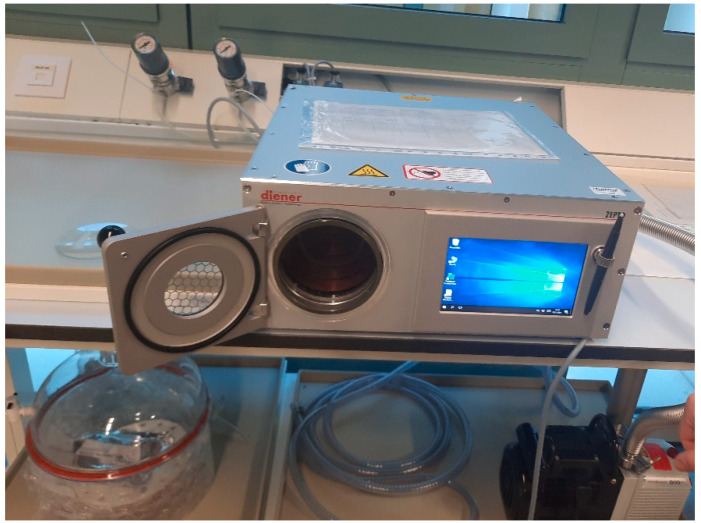
Low-pressure plasma apparatus used with argon and with polymerization treatment.

**Figure 6 materials-15-07487-f006:**
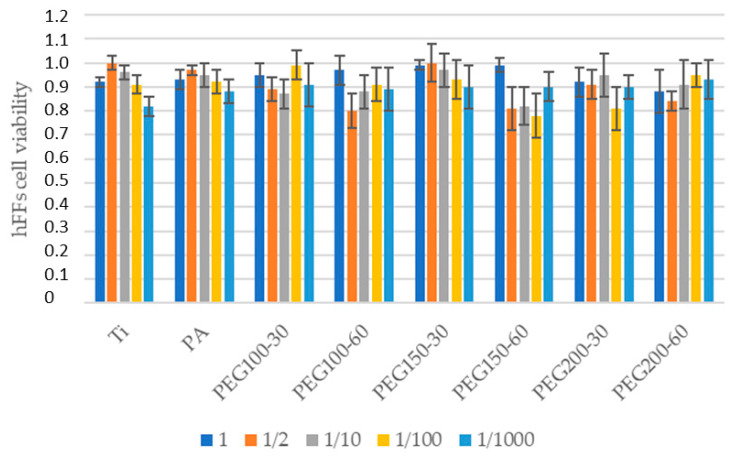
Cell viability of the hFFs for the different treatments and different dilutions.

**Figure 7 materials-15-07487-f007:**
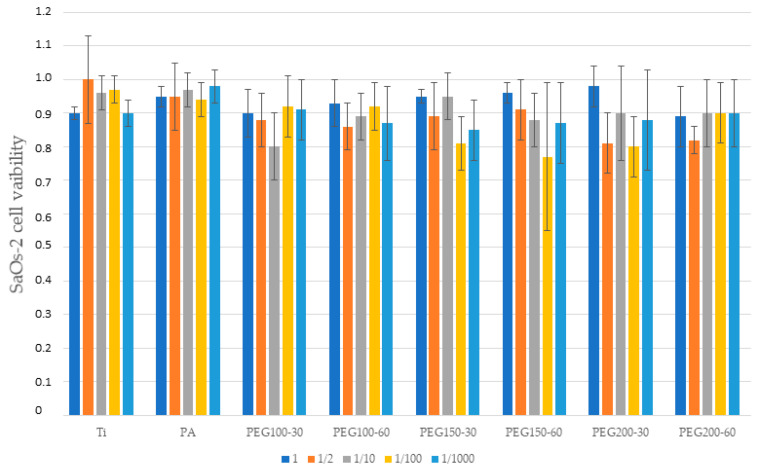
Cell viability of the SAOS-2 for the different treatments and different dilutions.

**Figure 8 materials-15-07487-f008:**
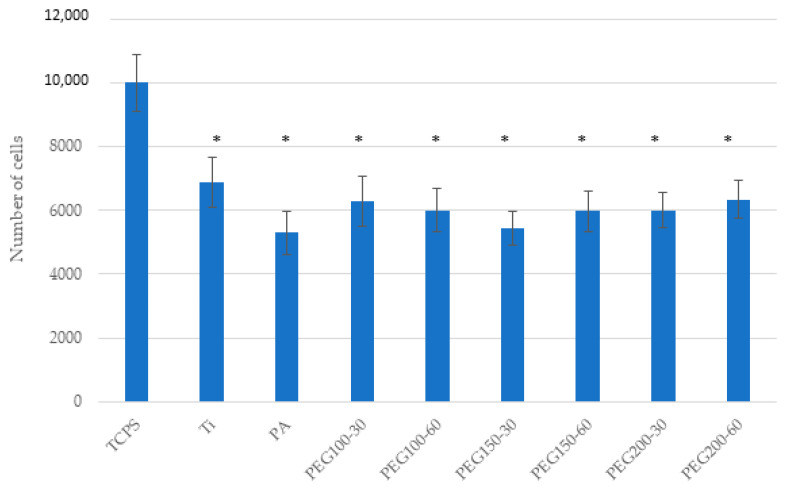
Cell adhesion of hFFs on the Ti, PA and PEG samples. Bars indicated with the same symbol have no statistically significant difference between them (*p* < 0.05).

**Figure 9 materials-15-07487-f009:**
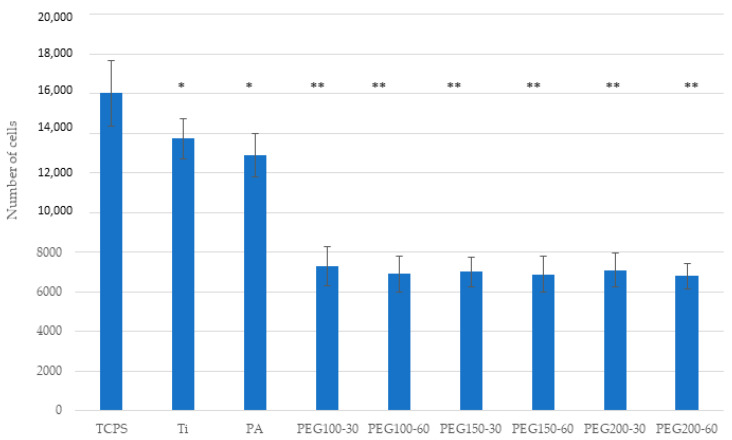
Cell adhesion of SaOS-2 on the Ti, PA and PEG samples. Bars indicated with the same symbol have no statistically significant difference between them (*p* < 0.05).

**Figure 10 materials-15-07487-f010:**
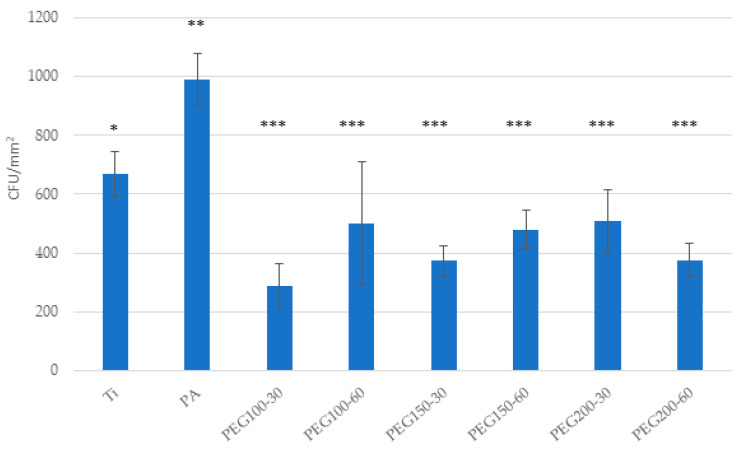
Bacterial adhesion on Ti, PA and PEG samples of *Spectrococcus sanguinis*. Asterisks indicated with the same symbol have no statistically significant difference between them (*p* > 0.05).

**Figure 11 materials-15-07487-f011:**
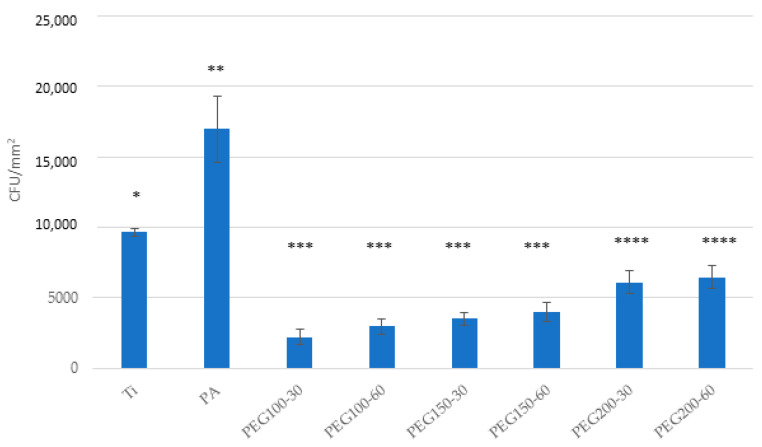
Bacterial adhesion on Ti, PA and PEG samples of
*Lactobacillus Salivarius*. Asterisks indicated with the same symbol have no statistically significant difference between them (*p* > 0.05).

**Table 1 materials-15-07487-t001:** Peak powers and times used in the PEG polymerization plasma treatments.

Sample	Peak Power (W)	Time (min)
PEG100-30	100	30
PEG100-60	100	60
PEG150-30	150	30
PEG150-60	150	60
PEG200-30	200	30
PEG200-60	200	60

**Table 2 materials-15-07487-t002:** Roughness of the different mini-implants.

Sample	R_a_ (μm)	P_c_ (cm^−1^)
Ti	0.33 ± 0.10	150.9 ± 69
PA	0.35 ± 0.20	153.4 ± 56
PEG100-30	0.36 ± 0.30	152.8 ± 59
PEG100-60	0.33 ± 0.21	146.9 ± 60
PEG150-30	0.43 ± 0.12	150.9 ± 69
PEG150-60	0.37 ± 0.09	150.9 ± 69
PEG200-30	0.39 ± 0.09	150.9 ± 69
PEG200-60	0.32 ± 0.12	150.9 ± 69

**Table 3 materials-15-07487-t003:** Contact angles for the different samples with PEG. Asterisks indicate the statistical difference significances.

Sample	Contact Angle (°)
PEG100-30	12.3 ± 0.9 *
PEG100-60	13.2 ± 1.3 *
PEG150-30	18.2 ± 1.2 **
PEG150-60	19.1 ± 0.8 **
PEG200-30	25.0 ± 2.2 ***
PEG200-60	25.2 ± 2.9 ***

**Table 4 materials-15-07487-t004:** Atomic concentration (in %) of the carbon, oxygen and titanium amount present on the Ti, and PA.

Sample	O 1s	C 1s	Ti 2p
Ti	55 ± 1	24 ± 1	20 ± 1
PA	63 ± 2	10 ± 1	26 ± 1
PEG100-30	45 ± 2	41 ± 1	2 ± 1
PEG100-60	54 ± 3	37 ± 2	5 ± 1
PEG150-30	42 ± 1	47 ± 1	8 ± 1
PEG150-60	45 ± 1	52 ± 1	8 ± 1
PEG200-30	48 ± 2	44 ± 2	7 ± 1
PEG200-60	40 ± 1	52 ± 3	6 ± 1

## Data Availability

The authors can provide details of the research requesting by letter and commenting on their needs.

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
