# Peer review of "Bacteriostatic Poly Ethylene Glycol Plasma Coatings for Orthodontic Titanium Mini-Implants"

_materials, 2022, doi:10.3390/ma15217487_

Round 1

Reviewer 1 Report

This paper is about “Bacteriostatic Poly Ethylene Glycol Plasma Coatings For Orthodontic Titanium Mini-Implants”. The title is suitable for the work but the work conceptually is not interesting. By the way, I have some suggestions to improve your work for further steps.

Line 19: “The roughness was determined by confocal microscopy” It is supposed to be atomic force microscopy? Where are the results.

Line 21: “fibroblasts and osteoblasts” better to change “fibroblast and osteoblast cells”

Keywords: “polyethileneglycol” typos error.

Lots of typos error and English polishing is necessary.

Line 71-72: these mechanisms are not make sence, explain in detail. What do these “chain compression and by acting as a barrier created by structured water associated with the PEG” mean?

Line 77: if you google “plasma polymerization PEG-like coatings titanium” you will find a lot of papers like. These papers are suggested to strengthen literature.

-       https://doi.org/10.1016/j.ultsonch.2019.104783

-       DOI: 10.1116/1.4913376

-       doi.org/10.1016/j.colsurfb.2017.01.042

-       https://doi.org/10.1111/lam.13418

-       /doi.org/10.1016/j.biomaterials.2007.06.028

-       https://doi.org/10.1016/j.surfcoat.2017.05.016

-       doi.org/10.1557/s43579-021-00071-7

-       https://doi.org/10.3389/fmicb.2021.619323

-       Biofunctional coating - postprint.pdf;sequence=1 (upc.edu)

-       dx.doi.org/10.1116/1.4913376

-        

Argon does not need to be written uppercase,

Line 92: “relaised” typos.

Line 104: rewrite the sentence.

Line 113-114: what is the impact of this line with 3 references?

Line 146: “Three samples for each condition were studied “ why here needs 2 refrences?

Line 147: “from [36]” this reporting format is not scientific.

The number of cells and bacteria in whole text should be written in correct format of power formula, correct it. “2·104cells”, “1·108 colony”

“Student’s t-tests” no need to be uppercase letter.

The data reports in Table 2, looks weird. I can not find significant difference between them except “PEG150-30”, “PEG200-30”

Line 172: rewrite it.

The bar chart figures are suggested to be drawn in Origin and more eye-catching.

Line 223: “the PEG-treated 223 samples reduce the contact angle by more than half” how contact angle decreases but

Line 228: “and of carbon” correct it.

Line 224: “is shown” and “in which 244 it has been studied” correct them.

 Line 245: “PEG coatings produce a decrease in the adhesion of the BSA protein that favors bacterial colonization”, so PEG favors bacterial colonization? Does it look in contrast with your paper?

Line 249-251: “The results of the titanium 249 treated with argon plasma show good values of biocompatibility and cell adhesion but do 250 not offer good results in bacterial colonization” why? What is the mechanism?

Line 259: “has decreased” past tense.

Line 260: “both cell types” correct it.

Line 263: “However, the titanium surface only activated by argon plasma causes an increase in bacterial colonization” again contradict!

Also, some grammatical errors exist in the text and need soft polishing.

Abstract should be reduced to 250 words maximum, which should be highlight work, objectives, work and result novelty

Most literatures are too old

Incorporate last 5 years papers of about 50%, the references are very old.

Homogenize the references format.

Author Response

Dear Reviewer,

Thanks for taking the time to review our manuscript and suggest to us to improve our work by providing a lot more detail. We have done so, and we are now submitting a manuscript that not only addresses the points you specifically raised but also many others that we have considered in order to deliver what we think is a much improved version of our work. This version includes more paragraphs, English grammar revisions in all main sections, new references. Thanks a lot. We are looking forward to your comments.

Sincerely,

This paper is about “Bacteriostatic Poly Ethylene Glycol Plasma Coatings For Orthodontic Titanium Mini-Implants”. The title is suitable for the work but the work conceptually is not interesting. By the way, I have some suggestions to improve your work for further steps.

This research work is the result of a European research project in which the focus is on the fight against peri-implantitis. This oral disease is one of the main priorities of the strategic document and the aim is to work on preventing these diseases with coatings or topography of implants and mini-implants to avoid bacterial colonisation. Plasma polymerisation coatings are one of the most promising methods and a Concerted Action of the European Union has been granted to promote these studies. I believe that in this new version of the paper, where we have taken into consideration your numerous contributions, it has gained in interest and understanding. I am grateful for the great work of the reviewer in improving the paper.

Line 19: “The roughness was determined by confocal microscopy” It is supposed to be atomic force microscopy? Where are the results.

This is a mistake; The text has been changed.

Line 21: “fibroblasts and osteoblasts” better to change “fibroblast and osteoblast cells”

Done

Keywords: “polyethileneglycol” typos error.

Corrected

Lots of typos error and English polishing is necessary.

Revised

Line 71-72: these mechanisms are not make sence, explain in detail. What do these “chain compression and by acting as a barrier created by structured water associated with the PEG” mean?

The sentence has been improved. Thank you for this comment.

Line 77: if you google “plasma polymerization PEG-like coatings titanium” you will find a lot of papers like. These papers are suggested to strengthen literature.

-       https://doi.org/10.1016/j.ultsonch.2019.104783

-       DOI: 10.1116/1.4913376

-       doi.org/10.1016/j.colsurfb.2017.01.042

-       https://doi.org/10.1111/lam.13418

-       /doi.org/10.1016/j.biomaterials.2007.06.028

-       https://doi.org/10.1016/j.surfcoat.2017.05.016

-       doi.org/10.1557/s43579-021-00071-7

-       https://doi.org/10.3389/fmicb.2021.619323

-       Biofunctional coating - postprint.pdf;sequence=1 (upc.edu)

-       dx.doi.org/10.1116/1.4913376

-       All references and new paragraphs have been introduced to complement the text, as well as new ideas and comments on the texts brought to our attention by the reviewer. 

Argon does not need to be written uppercase,

Changed Argon by argon

Line 92: “relaised” typos.

Corrected

Line 104: rewrite the sentence.

Done

Line 113-114: what is the impact of this line with 3 references?

The authors thought that some references explaining in more detail the experimentation and the results obtained could be interesting. This model of equipment has some very interesting characteristics and has a certain novelty. The reading of other works in which this equipment is used may be of interest.

Line 146: “Three samples for each condition were studied “ why here needs 2 refrences?

The reference number were in an incorrect position. The references show more details of the experimental procedure in case readers want to know more about it.

Line 147: “from [36]” this reporting format is not scientific.

Corrected

The number of cells and bacteria in whole text should be written in correct format of power formula, correct it. “2·104cells”, “1·108 colony”

Done.

“Student’s t-tests” no need to be uppercase letter.

Done

The data reports in Table 2, looks weird. I can not find significant difference between them except “PEG150-30”, “PEG200-30”

On the basis of the results of Table 2, no statistically significant differences can be observed in any case. The treatment does not affect the topography of the samples. These sentences have been introduced in the text

Line 172: rewrite it.

Done

The bar chart figures are suggested to be drawn in Origin and more eye-catching.

The figures have been improved

Line 223: “the PEG-treated 223 samples reduce the contact angle by more than half” how contact angle decreases but

The sentence has been improved.

Line 228: “and of carbon” correct it.

Done

Line 224: “is shown” and “in which 244 it has been studied” correct them.

Corrected

Line 249-251: “The results of the titanium 249 treated with argon plasma show good values of biocompatibility and cell adhesion but do 250 not offer good results in bacterial colonization” why? What is the mechanism?

The mechanism of bacterial adhesion depends on the wettability and hydrophilicity of the surface. The fact that the contact angles are very small prevents bacteria with a hydrophobic character, such as those common in the mouth, from adhering to the surface and therefore produces a decrease in bacterial colonization between strains of S. Sanguinis and L. Salivarius.

Line 259: “has decreased” past tense.

Done

Line 260: “both cell types” correct it.

Done

Line 263: “However, the titanium surface only activated by argon plasma causes an increase in bacterial colonization” again contradict!

In this case it does not have the bacteriostatic layer of the PEG that causes the inhibition of bacteriostatic adhesion and that is why the activation only with Argon generates the surface energy that causes bacterial adhesion.

Also, some grammatical errors exist in the text and need soft polishing.

Revised

Abstract should be reduced to 250 words maximum, which should be highlight work, objectives, work and result novelty.

Abstract has been improved and the words have been reduced

Most literatures are too old

Incorporate last 5 years papers of about 50%, the references are very old.

Fifteen new references have been introduced from 2015 to 2022.

Homogenize the references format.

Done

Reviewer 2 Report

The authors have conducted studies on coatings performed on micro-sized titanium implants and different characterization is done.  To this regard I have the following comments that need to be addressed before further consideration:

1. The abstract must contain significant results from the study and not just texts. 

2. The figures does not have appropriate labeling and the authors need to check the quality of each of the figure. 

3. Can the authors re-define their research gap and aim in the later part of the introduction. It needs revision. 

4. Figure 3 needs dimension check. 

5. There are several grammatical errors throughout the manuscript. 

6. Figure 4 should be replaced with schematic representation or can add an image. 

7. The equipment used for each characterization in the study needs explanation (make, model, measuring conditions, etc. ). Better to include a new sub-section for this.

8. How was the repeatability ensured in the experiments. How many set of experiments were performed in the study? 

Author Response

REVIEWER 2

Dear Reviewer,

Thanks for taking the time to review our manuscript and suggest to us to improve our work by providing a lot more detail. We have done so, and we are now submitting a manuscript that not only addresses the points you specifically raised but also many others that we have considered in order to deliver what we think is a much improved version of our work. This version includes more paragraphs, English grammar revisions in all main sections, new references. Thanks a lot. We are looking forward to your comments.

Sincerely,

The authors have conducted studies on coatings performed on micro-sized titanium implants and different characterization is done.  To this regard I have the following comments that need to be addressed before further consideration:

  1. The abstract must contain significant results from the study and not just texts. 

The abstract has been improved with results and reduced in extension

  1. The figures does not have appropriate labeling and the authors need to check the quality of each of the figure. 

The Figure legends have been improved

  1. Can the authors re-define their research gap and aim in the later part of the introduction. It needs revision. 

A new paragraph has been introduced in the text.

  1. Figure 3 needs dimension check. 

Done

  1. There are several grammatical errors throughout the manuscript. 

Revised

  1. Figure 4 should be replaced with schematic representation or can add an image. 

A new figure has been introduced according to the reviewer. The plasma equipment is in the text because one review comments its interest.

  1. The equipment used for each characterization in the study needs explanation (make, model, measuring conditions, etc. ). Better to include a new sub-section for this.

This section has been improved. More details have been added.

  1. How was the repeatability ensured in the experiments. How many set of experiments were performed in the study? 

This aspect has been introduced in the text (Materials and Methods)

Reviewer 3 Report

1.      The authors need to provide all of the emails after affiliation except for corresponding authors based on MDPI format.

2.      The abstract requires the addition of quantitative results.

3.      As the conclusion of your abstract, please provide a "take-home" message.

4.      Line 32, “we can confirm”, it is not scientific, make it into passive.

5.      Rearrange keywords alphabetically.

6.      Make the each of keywords with lowercase font following MDPI format, revise it.

7.      It is unclear whether the author's something new in this work. According to evaluation, several published studies by other researchers in the past adequately explain the issues you made in the present paper for PEG coating in dental implant. Please be careful to highlight in the introduction section anything really innovative in this work.

8.      In order to demonstrate the research gaps that the current study aims to address, previous studies linked to it need to be explained in the introduction part, including their work, their novelty, and their limitations.

9.      The authors need to explain the reasons for using titanium materials, it would be from biocompability and/or biomechanical aspect. It is a vital topic that authors must provide in the introduction and/or discussion section. Additionally, the MDPI's suggested reverence should be taken to substantiate this explanation as follows: Ammarullah, M. I.; Afif, I. Y.; Maula, M. I.; Winarni, T. I.; Tauviqirrahman, M.; Jamari, J. Tresca Stress Evaluation of Metal-on-UHMWPE Total Hip Arthroplasty during Peak Loading from Normal Walking Activity. Mater. Today Proc. 2022, 63, S143–6. https://doi.org/10.1016/j.matpr.2022.02.055

10.   In the materials and methods, the authors need to add additional illustrations as a form of figure that explains the workflow of the present study to make the reader easier to understand rather than only the dominant text as a present form.

Author Response

REVIEWER 3

Dear Reviewer,

Thanks for taking the time to review our manuscript and suggest to us to improve our work by providing a lot more detail. We have done so, and we are now submitting a manuscript that not only addresses the points you specifically raised but also many others that we have considered in order to deliver what we think is a much improved version of our work. This version includes more paragraphs, English grammar revisions in all main sections, new references. Thanks a lot. We are looking forward to your comments.

Sincerely,

Francisco-Javier Gil Mur

  1. The authors need to provide all of the emails after affiliation except for corresponding authors based on MDPI format.

Done

  1. The abstract requires the addition of quantitative results.

Done. The abstract has been improved according to the reviewer’s comments

  1. As the conclusion of your abstract, please provide a "take-home" message.

Conclusions have been improved following the suggestion of the reviewer.

  1. Line 32, “we can confirm”, it is not scientific, make it into passive.

Done

  1. Rearrange keywords alphabetically.

Done

  1. Make the each of keywords with lowercase font following MDPI format, revise it.

Done

  1. It is unclear whether the author's something new in this work. According to evaluation, several published studies by other researchers in the past adequately explain the issues you made in the present paper for PEG coating in dental implant. Please be careful to highlight in the introduction section anything really innovative in this work.

In this work, PEG coatings have been applied to titanium mini implants for orthodontics. This contribution makes real the ability to perform coatings on an implant without modifying the roughness. This fact is very important since the mini implant has to be fixed to the bone in order to be able to apply the alignment stresses. The good cellular behavior has been determined as well as demonstrating that the bacteria most typical of the oral cavity suffer a very important decrease in their adhesion to the substrate. This is the first time that this coating has been applied to an implant and the good adhesion to the surface and the maintenance of its bacteriostatic capacity have been confirmed. 

  1. In order to demonstrate the research gaps that the current study aims to address, previous studies linked to it need to be explained in the introduction part, including their work, their novelty, and their limitations.

According to the reviewer, new texts have been introduced in the introduction with new recent references. In addition, the innovation of the present article has been introduced in the text as well as the contribution in Orthodontics. In this work, PEG coatings have been applied to titanium mini implants for orthodontics. This contribution makes real the ability to perform coatings on an implant without modifying the roughness. This fact is very important since the mini implant has to be fixed to the bone in order to be able to apply the alignment stresses. The good cellular behavior has been determined as well as demonstrating that the bacteria most typical of the oral cavity suffer a very important decrease in their adhesion to the substrate. This is the first time that this coating has been applied to an implant and the good adhesion to the surface and the maintenance of its bacteriostatic capacity have been confirmed. 

  1. The authors need to explain the reasons for using titanium materials, it would be from biocompability and/or biomechanical aspect. It is a vital topic that authors must provide in the introduction and/or discussion section. Additionally, the MDPI's suggested reverence should be taken to substantiate this explanation as follows: Ammarullah, M. I.; Afif, I. Y.; Maula, M. I.; Winarni, T. I.; Tauviqirrahman, M.; Jamari, J. Tresca Stress Evaluation of Metal-on-UHMWPE Total Hip Arthroplasty during Peak Loading from Normal Walking Activity. Mater. Today Proc. 2022, 63, S143–6. https://doi.org/10.1016/j.matpr.2022.02.055

 This reference has been introduced in the text explaining the interest of titanium for biomechanical and biocompatibility properties.

  1. In the materials and methods, the authors need to add additional illustrations as a form of figure that explains the workflow of the present study to make the reader easier to understand rather than only the dominant text as a present form.

Done. New figure 4 is a schematic illustration of the materials and methods. Thank you for your suggestion.

Round 2

Reviewer 1 Report

The abstract still needs to be improved, the tense of the text is not uniform, and the writing needs to be modified. The abbreviation abstract is not allowed.

The figures 1,2 belong to this study or were adopted from somewhere else? If yes, move to results and discussion or merge with figure 3, if not, put the reference.

Line 70-73, 12 references are reported for 2 simple sentences, or in Line 76 the same. It is not professional. Distribute references with reasonable etiquette.

Line 249-251: “The results of the titanium treated with argon plasma show good biocompatibility and cell adhesion values but do not offer good results in bacterial colonization” why? What is the mechanism? I am still not satisfied with your answer.

Line 85: really this is not first time, I had attached some references before, please highlight your work and revise your claim “for the first time”.

Figure 4 is not suitable for the work. Super elementary.

Figure 5 has no message.

Line 125: “Data analysis was performed with”, what?

Line 141: why you did 3 times XPS for each sample? XPS is not a statistical test, what is the reason?

“A”rgon is still in the text.

Table 4, why the trends of C and O are different, if you claim XPS is a suitable method of quantitative surface element characterization, (which is not for C and O), please provide a scientific reason for this.

Lines 269-272 move to the introduction.

Author Response

Thank you very much for your attention. Your comments have been cosnidered and have improved the contribution. Thanks again. 

REVIEWER 1

The abstract still needs to be improved, the tense of the text is not uniform, and the writing needs to be modified. The abbreviation abstract is not allowed.

The abstract has been improved according to the reviewer. The abbreviations have been explained in the text.

The figures 1,2 belong to this study or were adopted from somewhere else? If yes, move to results and discussion or merge with figure 3, if not, put the reference.

The reference has been introduced.

Line 70-73, 12 references are reported for 2 simple sentences, or in Line 76 the same. It is not professional. Distribute references with reasonable etiquette.

The references have been selected. In the new version, the references have been reduced from 12 to 5.

Line 249-251: “The results of the titanium treated with argon plasma show good biocompatibility and cell adhesion values but do not offer good results in bacterial colonization” why? What is the mechanism? I am still not satisfied with your answer.

I have introduced a paragraph with some considerations that may help to understand the effect of wettability with bacteria. It is a very controversial aspect since it depends on many factors such as wettability, roughness, bacterial strains, the way they are cultivated, among the most important ones. The work of Mauguier et al. shows that their results are in the same direction as ours, observing that superhydrophilicity favors bacterial adhesion, which is the case with argon. This fact is attributed to the ease of protein and bacterial adsorption. It is also shown that nanotextured and PEG surfaces have a chemical configuration that impairs bacterial adhesion due to the difficulty of protein adsorption. New references have been added.

Line 85: really this is not first time, I had attached some references before, please highlight your work and revise your claim “for the first time”.

The paragraph has been changed according to the reviewer.

Figure 4 is not suitable for the work. Super elementary.

Figure 4 has been improved.

Figure 5 has no message.

A new figure legend has been added

Line 125: “Data analysis was performed with”, what?

The sentence has been completed

Line 141: why you did 3 times XPS for each sample? XPS is not a statistical test, what is the reason?

Three samples were run for each treatment to determine reproducibility and calculation of mean values. It was observed that the standard deviation is small and therefore the treatment is reproducible, observing the cleanliness of the material and the polymerization performed.

“A”rgon is still in the text.

Corrected

Table 4, why the trends of C and O are different, if you claim XPS is a suitable method of quantitative surface element characterization, (which is not for C and O), please provide a scientific reason for this.

As the reviewer says the XPS is not very sensitive from a quantitative point of view but gives feedback on oxygen and carbon levels to check surface cleanliness as well as carbon increases due to coating. These analyses are in agreement with other authors in similar research. New references have been incorporated.

Lines 269-272 move to the introduction.

Done

Reviewer 3 Report

Good job to the authors, but I have some issue after revision that needs to be addressed as follows:

1.      It's also important to provide more particular information on tools, such as the manufacturer, the country, and the specification.

2.      Error and tolerance of experimental tools used in this work are important information that needs to be explained in the manuscript. It is would use as a valuable discussion due to different results in the further study by other researcher.

3.      Outcomes must be compared to similar past research.

4.      What is the limitation of the present work? Please include it before the conclusion section.

5.      Please discuss the further research in the conclusion section.

6.      The reference should be enriched with literature from the last five years. Literature published by MDPI is strongly recommended.

7.      The authors occasionally created paragraphs in the entire document that were just one or two phrases long, which made the explanation difficult to understand. To make their explanation into a longer, more thorough paragraph, the authors should expand it. It is advised to use at least three sentences in a paragraph, with one serving as the primary sentence and the others as supporting phrases.

8.      Due to grammatical and language issues, the authors need to proofread the present work. This problem would use MDPI English editing service.

9.      Explain the potential study from computational simulation (in silico) study needs to be explained that bring several advantages such as faster results and lower cost compared to clinical study. The introduction and/or discussion part of an article should contain this crucial information. In addition, to support this explanation, the MDPI-suggested reference should be included as follows: Jamari, J.; Ammarullah, M. I.; Santoso, G.; Sugiharto, S.; Supriyono, T.; Heide, E. van der. In Silico Contact Pressure of Metal-on-Metal Total Hip Implant with Different Materials Subjected to Gait Loading. Metals (Basel). 2022, 12, 1241. https://doi.org/10.3390/met12081241

10.   Please verify that the authors followed the MDPI format exactly, edit the current form, and recheck in addition to the other issues that have been noted.

Author Response

Thank you very much. The authors have considered all the comments. Your suggestions and comments have improved the contribution. Thanks again. 

REVIEWER 3

Good job to the authors, but I have some issue after revision that needs to be addressed as follows:

  1. It's also important to provide more particular information on tools, such as the manufacturer, the country, and the specification.

Model, manufacturer, city and state have been introduced in the material and methods.

  1. Error and tolerance of experimental tools used in this work are important information that needs to be explained in the manuscript. It is would use as a valuable discussion due to different results in the further study by other researcher.

Done

  1. Outcomes must be compared to similar past research.

In the discussion have been introduced results of other authors and are compared with the results of this contribution.

  1. What is the limitation of the present work? Please include it before the conclusion section.

Done

  1. Please discuss the further research in the conclusion section.

Done

  1. The reference should be enriched with literature from the last five years. Literature published by MDPI is strongly recommended.

New references have been introduced. Many of them published by MDPI

  1. The authors occasionally created paragraphs in the entire document that were just one or two phrases long, which made the explanation difficult to understand. To make their explanation into a longer, more thorough paragraph, the authors should expand it. It is advised to use at least three sentences in a paragraph, with one serving as the primary sentence and the others as supporting phrases.

Revised

  1. Due to grammatical and language issues, the authors need to proofread the present work. This problem would use MDPI English editing service.

The text has been revised

  1. Explain the potential study from computational simulation (in silico) study needs to be explained that bring several advantages such as faster results and lower cost compared to clinical study. The introduction and/or discussion part of an article should contain this crucial information. In addition, to support this explanation, the MDPI-suggested reference should be included as follows: Jamari, J.; Ammarullah, M. I.; Santoso, G.; Sugiharto, S.; Supriyono, T.; Heide, E. van der. In Silico Contact Pressure of Metal-on-Metal Total Hip Implant with Different Materials Subjected to Gait Loading. Metals (Basel). 2022, 12, 1241. https://doi.org/10.3390/met12081241

 Done

  1. Please verify that the authors followed the MDPI format exactly, edit the current form, and recheck in addition to the other issues that have been noted.

The authors have revised the format according MDPI included the references.